# Jun Dimerization Protein 2 (JDP2) Increases p53 Transactivation by Decreasing MDM2

**DOI:** 10.3390/cancers16051000

**Published:** 2024-02-29

**Authors:** Kasey Price, William H. Yang, Leticia Cardoso, Chiung-Min Wang, Richard H. Yang, Wei-Hsiung Yang

**Affiliations:** Department of Biomedical Sciences, School of Medicine, Mercer University, Savannah, GA 31404, USA; price.kasey@gmail.com (K.P.); theyangbossman@gmail.com (W.H.Y.); leticiacdn@gmail.com (L.C.); chiungminw@gmail.com (C.-M.W.); thelegoman700@gmail.com (R.H.Y.)

**Keywords:** p53, JDP2, MDM2, ATF3, transcriptional activity

## Abstract

**Simple Summary:**

Cancer is the leading cause of death in the United States and worldwide. The current study demonstrates how JDP2, part of AP-1 protein complex, interacts with p53 tumor suppressor. The results show that JDP2 binds to p53, increases p53 transactivation, and helps stabilize p53. JDP2 also binds MDM2, the major negative regulator of p53. This suggests that JDP2 is a new regulator of p53-MDM2 pathway.

**Abstract:**

The AP-1 protein complex primarily consists of several proteins from the c-Fos, c-Jun, activating transcription factor (ATF), and Jun dimerization protein (JDP) families. JDP2 has been shown to interact with the cAMP response element (CRE) site present in many cis-elements of downstream target genes. JDP2 has also demonstrates important roles in cell-cycle regulation, cancer development and progression, inhibition of adipocyte differentiation, and the regulation of antibacterial immunity and bone homeostasis. JDP2 and ATF3 exhibit significant similarity in their C-terminal domains, sharing 60–65% identities. Previous studies have demonstrated that ATF3 is able to influence both the transcriptional activity and p53 stability via a p53-ATF3 interaction. While some studies have shown that JDP2 suppresses p53 transcriptional activity and in turn, p53 represses JDP2 promoter activity, the direct interaction between JDP2 and p53 and the regulatory role of JDP2 in p53 transactivation have not been explored. In the current study, we provide evidence, for the first time, that JDP2 interacts with p53 and regulates p53 transactivation. First, we demonstrated that JDP2 binds to p53 and the C-terminal domain of JDP2 is crucial for the interaction. Second, in p53-null H1299 cells, JDP2 shows a robust increase of p53 transactivation in the presence of p53 using p53 (14X)RE-Luc. Furthermore, JDP2 and ATF3 together additively enhance p53 transactivation in the presence of p53. While JDP2 can increase p53 transactivation in the presence of WT p53, JDP2 fails to enhance transactivation of hotspot mutant p53. Moreover, in CHX chase experiments, we showed that JDP2 slightly enhances p53 stability. Finally, our findings indicate that JDP2 has the ability to reverse MDM2-induced p53 repression, likely due to decreased levels of MDM2 by JDP2. In summary, our results provide evidence that JDP2 directly interacts with p53 and decreases MDM2 levels to enhance p53 transactivation, suggesting that JDP2 is a novel regulator of p53 and MDM2.

## 1. Introduction

Cancer is the leading cause of death in the United States and worldwide in the 21st century. According to the Cancer Statistics from the National Cancer Institute (https://www.cancer.gov/about-cancer/understanding/statistics) (accessed on 1 January 2024) and the American Cancer Society (https://www.cancer.org) (accessed on 1 January 2024), the cancer mortality rate of human cancers is around 10–25%. Among the proteins which have been studied in the cancer field, p53 is the major tumor suppressor and cellular regulator studied most in the world and recognized as the 1993 Molecule of the Year. Mutations of p53 have been found in more than 50% of human cancers, suggesting p53’s significant role in responding to DNA damage [1]. Regulation of p53 has been shown in many different pathways and mechanisms with the negative action of MDM2 and MDM4 (MDMX) being the most important [2]. Extensive research has identified numerous p53 downstream target genes, such as p21 (CDN1A), PUMA (BBC3), GADD45, BAX, MDM2, and others [3,4,5,6]. As a tumor suppressor and transcription factor, p53 has been regulated by many regulatory processes including post-translational modifications (PTMs), such as acetylation [7], phosphorylation [8], methylation [9], and SUMOylation [10]. These PTMs are mainly located in the C-terminal domain of p53. Recent studies have demonstrated that p53 is involved in ferroptosis, a form of iron-dependent, non-apoptotic cell death [11]. Overall, p53 has broad effects on cellular process, such as the apoptosis regulatory pathway [12], maintenance of DNA integrity and repair function [13], autophagy signaling pathways [14], anabolism and catabolism pathways [15], cell cycle arrest and function [16], control of translation mechanism [17], and feedback loops such as the MDM2-p53 pathway [18].

The AP-1 protein complex primarily consists of several proteins from the c-Fos, c-Jun, activating transcription factor (ATF), and Jun dimerization protein (JDP) families. Like many other transcription factor complexes, AP-1 complex controls gene expression in response to several cellular signals. Many studies have supported the fact that AP-1 complexes are important in numerous tumorigenic processes [19]. Unlike c-Jun and c-Fos, JDP2 inhibits AP-1-mediated transactivation by recruiting histone deacetylase 3 (HDAC3) to the promoter areas, such as ATF3 promoter [20,21]. JDP2 has been shown to interact with the cAMP response element (CRE) and TPA response element (TRE) sites present in various cis-elements of target genes [22]. A study suggests that JDP2 can bind to histone directly as well [23]. Furthermore, many studies have demonstrated that JDP2 has broad functions in histone methyltransferase inhibition, histone acetyltransferase inhibition, and nucleosome assembly [24]. Generally, JDP2 has effects on numerous biological and cellular processes, including the inhibition of adipocyte differentiation and maturation [25], senescence regulation [26], activation of skeletal muscle differentiation [27], mediation of bone differentiation [28], regulation of cardiac function [29], interaction with and regulation of progesterone receptor in reproduction [30,31], maintenance of Epstein–Barr virus (EBV) latency [32], control of bone homeostasis, antibacterial immunity, and contribution to metastatic spread [33,34].

ATF3, consisting of 181 amino acids, and JDP2, consisting of 168 amino acids, share significant similarities in their C-terminal domains (60–65% identities). JDP2 plays an important role in regulating the activity of cell-cycle regulators, such as p16, cyclin A2, and cyclin E2, resulting in the initiating cell-cycle arrest via the RB-p16 and p53-p21 pathways [26,35]. A previous study has shown that JDP2 induces p16 and Arf by transducing signals from oxidative stress, resulting in cell-cycle arrest via both p16-pRb and p53-ARF pathways [36]. Moreover, a study indicates that the transcriptional activity of the mouse JDP2 promoter is inhibited by p53 [37]. Several studies have also demonstrated that JDP2 plays a role in both cancer development and progression. For example, JDP2 inhibits cell transformation by acting as a tumor suppressor in severe combined immune-deficient mice [38]. Decreased expression of JDP2 is associated with tumor metastasis and an unfavorable prognosis factor in patients with pancreatic cancers [39]. Interestingly, increased JDP2 gene activity is observed in head and neck cell lines [40]. A mouse study shows that JDP2 is able to increase the development of liver cancers [41]. Overall, these findings demonstrate the diverse role of JDP2 in many biological processes, especially cancer development.

However, the impact of JDP2 on the function and activity of p53 has not been fully described. Therefore, in this study, we assessed the role of JDP2 in regulating p53 transactivation. (Part of the current study was previously presented at the American Association for Cancer Research (AACR) annual meeting in 2022 with reprinted/adapted with permission from ref. [42], 2022, AACR). (Part of the current study was previously presented at the Endocrine Society annual meeting in 2022 with reprinted/adapted with permission from ref. [43], 2022, Journal of the Endocrine Society).

## 2. Materials and Methods

### 2.1. Reagents and Chemicals

The cell culture medium and cell culture reagents were purchased from Thermo Fisher Scientific (Walthan, MA, USA). Cycloheximide (CHX) was purchased from Sigma-Aldrich Inc. (St. Louis, MO, USA). Dual-Luciferase Reporter Assay System was purchased from Promega (Madison, WI, USA). Antibodies for p53 (sc-126, 1:3500), ATF3 (sc-81189, 1:2800), JDP2 (sc-23456, 1: 3000), MDM2 (sc-965, 1:3000), p21 (sc-6246, 1:3000), Nur77 (sc-365113, 1:3000), and β-actin (sc-81178, 1:6000) were purchased from Santa Cruz Biotechnology Inc. (Santa Cruz, CA, USA).

### 2.2. DNA Constructs

Human p53-pcDNA4 expression plasmids (with or without HIS tag) [44], human JDP2 WT and truncated expression plasmids, and human MDM2 expression plasmids used in this study were created using PCR-based In-Fusion technology (In-Fusion HD Cloning kit, Takara Bio, San Jose, CA, USA) in the Yang lab. Human ATF3-pcDNA3 expression plasmid was generated in the Yang lab as described previously [45]. The p53(14X)RE-Luc-pGL2 plasmid was kindly provided by Dr. Douglas Boyd (MD Anderson cancer Center, Houston, TX, USA) and Dr. Chunhong Yan (Georgia Cancer Center, Augusta University, Augusta, GA). Human p21 promoter(−2347 to +354)-Luc-pGL3 expression plasmid was generated in the Yang lab using PCR-based In-Fusion technology (In-Fusion HD Cloning kit, Takara Bio, San Jose, CA, USA). All DNA plasmid constructs were regularly verified by Sanger nucleotide sequencing.

### 2.3. Cell Culture

Cell culture was performed according to established protocols [46]. H1299 (CRL-5803) and MCF7 (HTB-22) cells were purchased from the American Type Culture Collection (ATCC) (Manassas, VA, USA). Cells were maintained in Dulbecco’s Modified Eagle Medium (DMEM), supplemented with 10% Fetal Bovine Serum (FBS) and 1% Penicilliin/Streptomycin antibiotics (Life Technologies, Grand Island, NY, USA) in a humidified incubator (5% CO_2_ at 37 °C) and maintained for a duration of less than six months. A PCR Detection Kit (Millipore-Sigma, Burlington, MA, USA) was used to verify the absence of Mycoplasma contamination.

### 2.4. Transient Transfection

The transfection of specific expression plasmids to cells was conducted by using the Fugene HD Transfection Reagent (Roche, Madison, WI, USA). After 48 h post transfection, the cells were harvested and lysed, and the lysate was subsequently used in luciferase reporter assays or Western blot analysis.

### 2.5. p53 (14X)RE Luciferase Reporter Assay

The reporter assays were performed following the established procedures [46]. Cells were plated in 24-well plates and cultured overnight before transfection. Cells were then transfected with the p53 (14X)RE-Luc reporter plasmid and the internal control pRL-TK plasmid (encoding Renilla luciferase activity). In addition, cells were either co-transfected with JDP2 or ATF3 expression plasmids. After 48 h post transfection, cells were collected and lysed with a passive lysis buffer (Promega, Madison, WI, USA). Luminescence was quantified using the Dual-Luciferase Reporter Assay System Kit (cat#E1960, Promega, Madison, WI, USA) and a luminometer (Turner Designs, Sunnyvale, CA, USA), following the manufacture’s instructions. The p53 (14X)RE-Luc luciferase activity was normalized by calculating the ratio to Renilla luciferase activity. The relative luciferase activity (LUC) was determined as a fold change compared to the control groups. Each experiment was conducted at least three times with triplicate samples.

### 2.6. Western Blot Analysis

The Western blot analysis was performed following the procedure outlined in a previous report [46]. After 48 h post transfection, cells were rinsed with chilled PBS and then lysed using chilled 1× RIPA buffer supplemented with phosphatase inhibitors and protease inhibitors. The protein content in the high-speed supernatant was measured using the BCA Protein Assay Kit (Pierce/Thermo Scientific, Rockford, IL, USA). Approximately 40 μg of the protein lysates were loaded and subjected to separation on 10% polyacrylamide-SDS gels. After electrophoresis, the proteins were transferred to a polyvinylidene difluoride (PVDF) membrane (Bio-Rad, Hercules, CA, USA) using wet electrophoretic transfer. The membranes wet were then blocked with 5% nonfat dry milk and then probed with specific primary antibodies, followed by specific secondary antibodies in TBST (0.1% Tween 20/TBS) buffer. Blots were visualized using the Supersignal West Dura Extended Duration Substrate Kit from Pierce Inc. (Rockford, IL, USA). The intensity of the protein band was quantified by using the ImageJ software (https://imagej.net/nih-image/, accessed on 1 January 2024) (NIH, Bethesda, MD, USA).

### 2.7. Statistical Analysis

Statistical analysis and comparison were performed using a two-tailed *t* test or a Mann–Whitney *U* test to ascertain the statistical significance between the groups. A *p* < 0.05 indicates statistical significance between the groups.

## 3. Results

### 3.1. JDP2 and ATF3 Increase p53 Transactivation

Since ATF3 and JDP2 share significant similarities in their C-terminal domains, we first assessed the impact of JDP2 and ATF3 on p53 transactivation using the p53 (14X)RE-Luc system. We used H1299 cells (human lung epithelial cancer cells) since they express minimal to no endogenous p53 proteins. As expected, the presence of WT p53 alone led to a significant increase in the p53 (14X)RE-Luc system, ranging from 50 to 2000 folds in various experiments. As shown in Figure 1A, the induction of WT ATF3 in the presence of p53 led to a significant increase in p53 transactivation, consistent with the previous study [47]. Interestingly, as shown in Figure 1B, WT JDP2 in the presence of p53 also demonstrated a significant increase in p53 transactivation in H1299 cells. To validate the effect of JDP2 in a different cellular context, we transfected MCF7 cells (which express WT p53 endogenously) with WT JDP2 and p53 (14X)RE-Luc plasmids. As shown in Figure 1C, JDP2 induces a 2-fold increase in p53 transactivation in MCF7 cells.

### 3.2. p53 Binds the C-Terminal Domain of JDP2

Since ATF3 is shown to bind to p53 [47] and JDP2 can enhance p53 transactivation in the presence of p53, we next investigated whether JDP2 is also able to bind to p53. As shown in Figure 2A, JDP2 is able to bind to p53 in immunoprecipitation assay. Moreover, as shown in Figure 2B, both ATF3 and JDP2 are able to bind to p53 when both ATF3 and JDP2 were examined in the same experiment. Since ATF3 has been shown to bind to p53 via its C-terminal domain [47], we next tested whether the C-terminal domain of JDP2 is essential for JDP2-p53 interaction. As shown in Figure 2C, the C-terminal domain (amino acids 141–168) is critical for the binding of JDP2 to p53. These results suggest that JDP2 is a novel binding partner for p53, and that both ATF3 and JDP2 use the C-terminal domain to interact with p53.

### 3.3. JDP2 and ATF3 Additively Enhance p53 Transactivation and the C-Terminal Domain of JDP2 Is Required for Enhancement of p53 Transactivation

Since ATF3 and JDP2 independently increase p53 transactivation (Figure 1 and Figure 2), we next investigated whether their combined presence results in an additive or synergistic effect on p53 transactivation. As shown in Figure 3A, the combined presence of ATF3 and JDP2 results in an additive effect in p53 transactivation. Since thousands of p53 mutations have been reported in human cancers, we next explored whether JDP2 can increase p53 transactivation in the presence of mutant p53. As shown in Figure 3B, while JDP2 effectively increases the transactivation of WT p53, JDP2 fails to activate the transactivation of R175H (a hotspot human p53 conformational mutation), R273H (a hotspot human p53 contact mutation), and G245S (a hotspot human p53 conformational mutation). Since JDP2 binds to p53 via its C-terminal domain, we tested whether the C-terminal domain of JDP2 is necessary for p53 transactivation. As shown in Figure 3C, the C-terminal domain (especially amino acids 120–168) is essential for increasing p53 transactivation. These results suggest that JDP2 increases WT p53 transactivation via its C-terminal domain.

### 3.4. JDP2 Slightly Increases p53 Stability in H1299 Cells

Since JDP2 binds to p53 and increases p53 transactivation (Figure 1 and Figure 2), and because ATF3 binding to p53 increases p53 stability [47], we investigated whether JDP2 affects p53 stability. To assess this question, cycloheximide (CHX) experiments were performed by introducing CHX to H1299 cells expressing either p53 alone or both p53 and JDP2 at various time point. As shown in Figure 4, the levels of p53 in the presence of JDP2 were slightly increased compared to those without p53 (approximately a 25% difference at 4-h time point). This result indicates that JDP2 can slightly stabilize p53 in H1299 cells.

### 3.5. JDP2 Decreases MDM2 Levels

Since JDP2 binds to p53, increases p53 transactivation (Figure 1, Figure 2 and Figure 3), and slightly stabilizes p53 (Figure 4), we then investigated whether JDP2 is able to bind to MDM2 and affects the levels of MDM2, the main negative regulator of p53. As shown in Figure 5A, JDP2 is able to bind to MDM2 in Ni-bead pulldown assay. To assess whether JDP2 affects the levels of MDM2, we transfected MDM2 expression plasmid and various doses of JDP2 expression plasmids to MCF7. As shown in Figure 5B, JDP2 dose-dependently decreases the levels of MDM2. In order to confirm JDP2’s effect on MDM2, we examined whether this effect can be observed for endogenous MDM2 in a different cellular context. We also tested Nur77 in the same experiments because Nur77 has been shown to repress the expression of MDM2 [48]. As shown in Figure 5C, JDP2 dose-dependently decreases the levels of endogenous MDM2 in H1299 cells. As expected, Nur77 also decreases MDM2 levels. Since JDP2 decreases MDM2 levels, we examined whether the C-terminal domain of JDP2 is required for MDM2 repression. As shown in Figure 5D, the C-terminal domain (amino acids 140–168) is indeed essential for JDP2-mediated MDM2 repression. These results suggest that JDP2 has the ability to bind to MDM2 and decrease the levels of MDM2.

### 3.6. While MDM2 Decreases p53 Transactivation, JDP2 Dose-Dependently Abolishes MDM2-Mediated p53 Repression

Since MDM2 is the major negative regulator of p53 and JDP2 decreases MDM2 level (Figure 5), we examined whether JDP2 can rescue MDM2-mediated p53 repression using p53 (14X)RE-Luc system. H1299 cells were co-transfected with a p53 (14X)RE-Luc reporter plasmid and a pRL-TK Renilla control plasmid, along with or without p53, MDM2, and JDP2 (different doses) expression plasmids. As shown in Figure 6, while MDM2 significantly decreases p53 transactivation, JDP2 dose-dependently reverses MDM2-mediated p53 repression. This result suggests that JDP2 reduces MDM2-mediated p53 repression by reducing MDM2 level.

### 3.7. JDP2 Increases p21 Promoter Activity and p21 Protein Level in the Presence of p53

Since p21 serves as the downstream gene of p53, we examined how JDP2 influences p21 promoter activity in the presence of p53. The binding site of p53 on p21 promoter is located from −2253 to −2232 (GAACATGTCCCAACATGTTG). H1299 cells were co-transfected with a p21 promoter(−2347 to +354)-Luc-pGL3 reporter plasmid and a pRL-TK Renilla control plasmid, and with or without p53 and JDP2 expression plasmids. As shown in Figure 7A, JDP2 demonstrated a significant increase in p21 promoter activity in the presence of p53, which suggests increased p53 transactivation in the presence of JDP2. To validate the effect of JDP2 in p21 protein levels in the presence of p53, we transfected cells with JDP2 expression plasmid, or p53 expression plasmid, or combination of JDP2 and p53 expression plasmids. As shown in Figure 7B, JDP2 demonstrated an increase in p21 protein level in the presence of p53. Overall, these findings suggest that JDP2 is a novel regulator in up-regulating p53 transactivation in a p53’s downstream gene activity.

## 4. Discussion

Transcription factor networks are very important for normal biological and cellular process and development, and are often disrupted in human disorders [46,49,50,51,52,53,54]. As a transcription factor and a tumor suppressor, p53 networks have been shown to regulate various functions including apoptosis, senescence, cell cycle regulation, DNA repair, metabolism, redox control, genomic stability, and differentiation and maturation [55,56]. Mutations or deletions of the TP53 gene have been found in more than 50% of human cancers [1,57]. In this study, we present a fascinating discovery that JDP2 binds to p53 and enhances p53 transactivation likely by decreasing the levels of MDM2.

ATF3 and JDP2 exhibit significant similarity in their C-terminal domains [58]. By using a truncation strategy, we identified that the region between amino acids 141 and 168 is crucial for JDP2 binding to p53 (Figure 2). This finding is consistent with the previous study [47] that the C-terminal domain is essential for ATF3 binding to p53. Therefore, the data from this study and the previous study [46] demonstrates that the C-terminal domains of both ATF3 and JDP2 play an essential role in binding p53 and the regulation of p53 networks.

Over the last two decades, the relationship between ATF3 and p53 has been expanded, such as (1) ATF3 plays a role in regulating the stability of p53 [59], (2) p53 transcriptionally upregulates the ATF3 gene promoter [60], (3) SUMOylation of ATF3 modifies its transcriptional activity in the regulation of TP53 gene [45], (4) ATF3 activates p53 by blocking its ubiquitination [47], and (5) ATF3 suppresses the oncogenic function of mutant p53 proteins [61]. Since JDP2 shares significant homology in the C-terminal domain with ATF3, it is likely that JDP2 exerts a similar impact on p53. Our current study provides evidence that JDP2 is able to bind to p53 through its C-terminal domain. Moreover, our findings suggest an additive effect of ATF3 and JDP2 on p53 transactivation. However, JDP2 is unable to transactivate the conformational (R175H and G245S) and contact (R273H) p53 mutants, suggesting that JDP2 cannot restore mutant p53’s trans-repression. Therefore, future studies are needed to dissect whether JDP2 can directly bind to human p53 mutants (more than thousands) found in major cancers.

Over the past decade, many studies have demonstrated the functions of JDP2, such as: (1) serving as a histone chaperone in cellular senescence and aging [62], (2) functioning as a cellular survival protein critical for normal cellular activities [63], and (3) inhibiting UV-mediated apoptosis via the down-regulation of p53 [64]. More recently, JDP2 has played roles in atrial fibrillation, cardiac remodeling, and heart failure. In addition, more studies have demonstrated that deficiencies in both JDP2 and ATF3 contribute to maladaptive cardiac remodeling while preserving the cardiac function [65]. Interestingly, in mouse models, it has been shown that p53 negatively regulates JDP2 [37], and JDP2, in turn, down-regulates TrP53 transcription to promote leukemogenesis, suggesting a regulatory loop between p53 and JDP2. Therefore, the present study provides the interaction between JDP2 and p53, suggesting that future studies are needed to dissect the underlying mechanisms of JDP2-p53 regulatory loop in details.

MDM2 is a major p53 negative regulator, as it binds to p53 and increases p53 degradation [66]. In this study, our results reveal that in addition to binding to p53 and enhancing p53 transactivation, JDP2 is able to reverse MDM2-mediated p53 repression in p53 transactivation study. Our results also demonstrate that JDP2 reduces the levels of MDM2 in a dose-dependent manner, suggesting that reducing the levels of MDM2 is the likely the mechanism that JDP2 increases p53 transactivation. Our result also indicates that JDP2 is capable of stabilizing p53 but the effect is fairly minimal. Therefore, future studies are indeed necessary to dissect the potential regulatory mechanisms between JDP2 and MDM2 in the regulation of p53.

Post-translational modifications (PTMs), including acetylation, methylation, phosphorylation, glycosylation, SUMOylation, ADP-ribosylation, O-GlcNAcylation, and ubiquitination, substantially impact a wide range of cellular activities [67,68]. ATF3 and JDP2 proteins are regulated by PTMs, such as phosphorylation of T148 in JDP2 [69]. In the previous study, our laboratory identified K42 as the major SUMOylation site in ATF3 [45]. While JDP2 has been reported to be regulated by SUMOylation from our previous study [70], the specific SUMOylation site(s) remain unidentified. Since both JDP2 and ATF3 can enhance p53 transactivation, future studies are necessary to explore whether PTMs play an important role in modulating JDP2- and ATF3-mediated p53 transactivation.

In conclusion, this study establishes JDP2 as a novel regulator of p53 transactivation, and the C-terminal domain of JDP2 plays a crucial role in JDP2-mediated p53 transactivation.

## 5. Conclusions

In summary, we propose an intriguing discovery that JDP2 increases p53 transactivation, likely due to decreasing MDM2 levels. This study suggests that JDP2 is a novel regulator of p53-MDM2 pathway. Moreover, the C-terminal domain of JDP2 is essential to execute JDP2’s function in the p53-MDM2 pathway. Therefore, our results provide a new layer of information to how JDP2 regulates p53-MDM2 pathway.

## Figures and Tables

**Figure 1 cancers-16-01000-f001:**
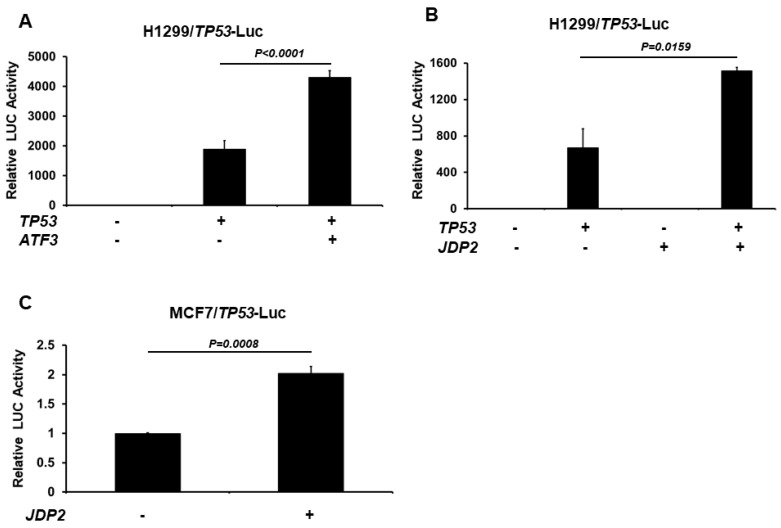
ATF3 and JDP2 increase p53 transactivation. Luciferase assays were performed to analyze p53 transactivation in H1299 cells (**A**,**B**) transfected either with or without WT p53 expression plasmids, in the absence or presence of ATF3 expression plasmid (**A**) or JDP2 expression plasmid (**B**). The luciferase assays were performed to analyze p53 transactivation in MCF7 cells (**C**) transfected either with or without JDP2 expression plasmids. After 48 h post transfection, luciferase activities were measured using the Dual Luciferase Reporter System and normalized to the control Renilla activity. The relative luciferase (LUC) activity was calculated and plotted. A *p* < 0.05 indicates statistical significance between groups.

**Figure 2 cancers-16-01000-f002:**
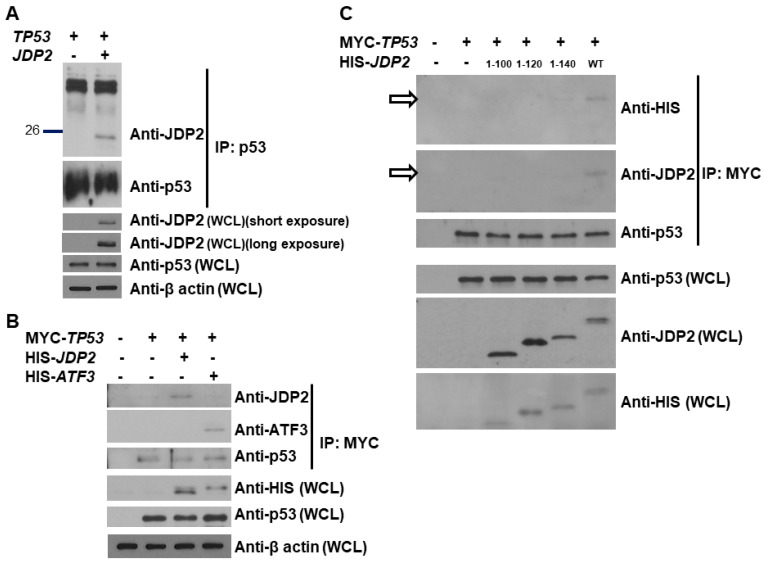
p53 interacts with the C-terminal domain of JDP2. (**A**) H1299 cells were transfected with a p53 expression plasmid, either with or without a JDP2 expression plasmid. After 48 h post transfection, whole cell lysates were immunoprecipitated using an anti-p53 antibody. The immunoprecipitated fractions were then subjected to Western blot analysis using anti p53 and anti-JDP2 antibodies. Protein levels of p53, JDP2, and β-actin in the whole cell lysates were validated by anti-p53, anti-JDP2, and anti-β-actin immunoblotting, respectively. (**B**) H1299 cells were transfected with a MYC-tagged p53 expression plasmid, with or without HIS-tagged JDP2 or ATF3 expression plasmids. After 48 h post transfection, whole cell lysates were immunoprecipitated using an anti-MYC antibody. The immunoprecipitated fractions were then subjected to Western blot analysis using anti p53, anti-ATF3, and anti-JDP2 antibodies. Protein levels of p53, JDP2, and β-actin in the whole cell lysates were validated by anti-p53, anti-HIS, and anti-β-actin immunoblotting, respectively. (**C**) H1299 cells were transfected with a MYC-tagged p53 expression plasmid, with or without HIS-tagged WT or truncated JDP2 expression plasmids. After 48 h post transfection, whole cell lysates were immunoprecipitated using an anti-MYC antibody. The immunoprecipitated fractions were then subjected to Western blot analysis using anti p53 and anti-HIS antibodies. Protein levels of p53, JDP2 (both WT and truncated forms) in the whole cell lysates were validated by anti-p53, anti-HIS, and anti-JDP2 immunoblotting, respectively. The original western blot figures can be found in Appendix A.

**Figure 3 cancers-16-01000-f003:**
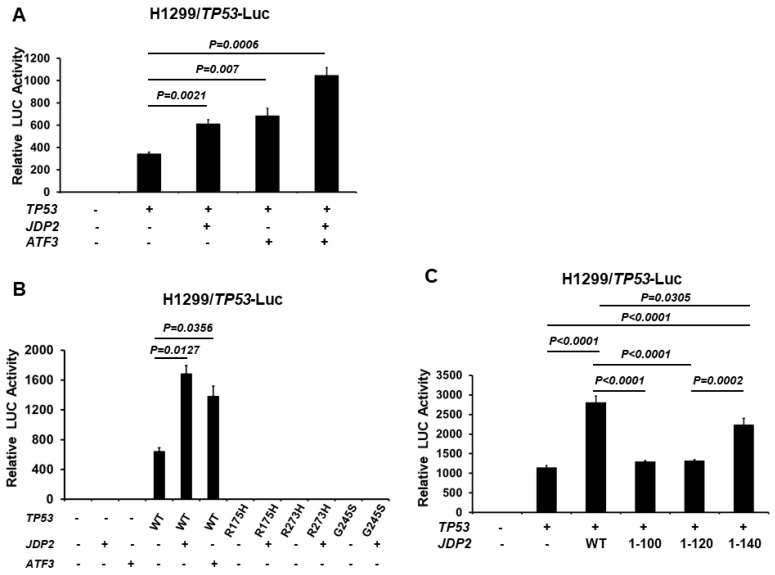
JDP2 and ATF3 additively increase p53 transactivation and the C-terminal domain of JDP2 is required for enhancement of p53 transactivation. (**A**) H1299 cells in a 24-well plate were co-transfected with the p53 (14X)RE-Luc reporter plasmid, pRL-TK Renilla control plasmid, and p53 expression plasmid, either with or without JDP2 or ATF3 or combination of JDP2 and ATF3 expression plasmids using Fugene HD transfection reagent. (**B**) H1299 cells in a 24-well plate were co-transfected with the p53 (14X)RE-Luc reporter plasmid, pRL-TK Renilla control plasmid, with or without JDP2 expression plasmid, and either WT, R175H, R273H, or G245S p53 expression plasmid. (**C**) H1299 cells in a 24-well plate were co-transfected with the p53 (14X)RE-Luc reporter plasmid, pRL-TK Renilla control plasmid, p53 expression plasmid, and either WT or truncated JDP2 expression plasmids. After 48 h post transfection, luciferase activities were measured using the Dual Luciferase Reporter System and normalized to the control Renilla activity. The relative luciferase (LUC) activity was calculated and plotted. A *p* < 0.05 indicates statistical significance between groups.

**Figure 4 cancers-16-01000-f004:**
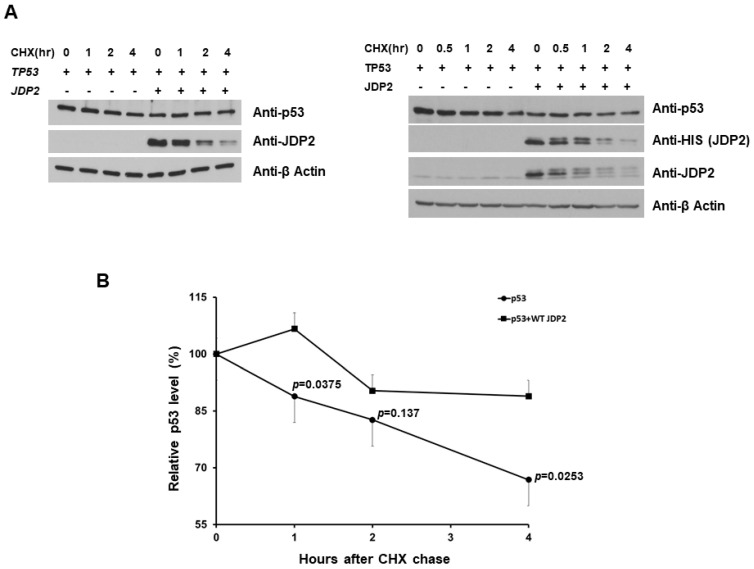
JDP2 slightly increases p53 stability in CHX experiments. CHX was introduced to H1299 cells which were transfected either p53 alone or combination of p53 and JDP2 at various time points. After 48 h post transfection, cells were harvested and the cell lysates were objected to anti-p53, anti-JDP2, and anti-β-actin immunoblotting (**A**). The protein levels of p53 and JDP2 were quantified by imageJ and plotted (**B**). A *p* < 0.05 indicates statistical significance between groups. The original western blot figures can be found in Appendix A.

**Figure 5 cancers-16-01000-f005:**
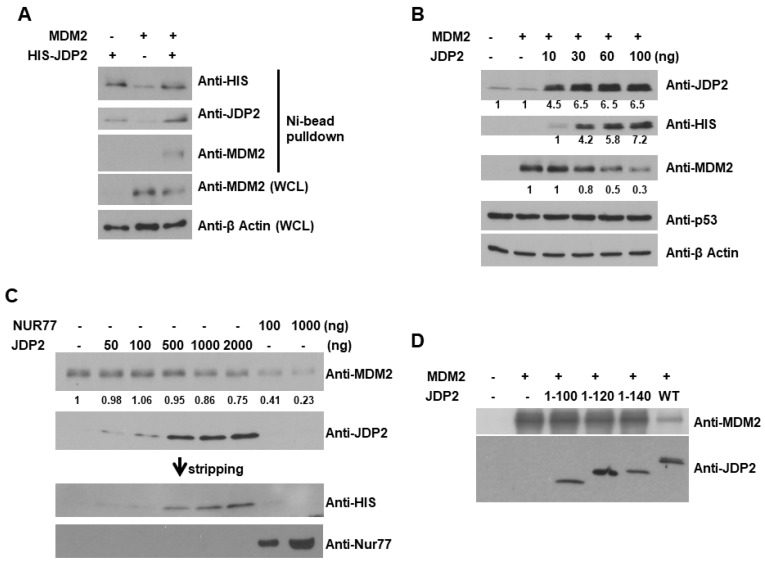
JDP2 decreases MDM2 levels. (**A**) H1299 cells were transfected with a HIS-tagged JDP2 expression plasmid, or MDM2 expression plasmid, or combination of HIS-tagged JDP2 and MDM2 expression plasmids by Fugene HD transfection method. After 48 h post transfection, whole cell lysates were precipitated using a Ni-beads to pulldown HIS-tagged JDP2. The precipitated fractions were then subjected to Western blot analysis using anti-JDP2, anti-HIS, and anti-MDM2 antibody. Protein levels of MDM2 and β-actin in the whole cell lysates were validated by anti-MDM2 and anti- β-actin immunoblotting, respectively. (**B**) MCF7 cells in a 6-well plate were co-transfected with MDM2 expression plasmid and with various doses of JDP2 expression plasmids by Fugene HD transfection method. After 48 h post transfection, cells were harvested and the whole cell lysates were subjected to anti-MDM2, anti-JDP2, anti-HIS, anti-p53, and anti-β-actin immunoblotting. (**C**) H1299 cells were transfected with JDP2 or Nur77 expression plasmid by the Fugene HD transfection method. After 48 h post transfection, cells were harvested and the whole cell lysates were subjected to anti-MDM2 (endogenous), anti-JDP2, anti-HIS (after stripping membrane), and anti-Nur77 immunoblotting. (**D**) MCF7 cells in a 6-well plate were co-transfected with MDM2 and different lengths of JDP2 expression plasmids by the Fugene HD transfection method. After 48 h post transfection, cells were harvested and the whole cell lysates were subjected to anti-MDM2 and anti-JDP2 immunoblotting. The original western blot figures can be found in Appendix A.

**Figure 6 cancers-16-01000-f006:**
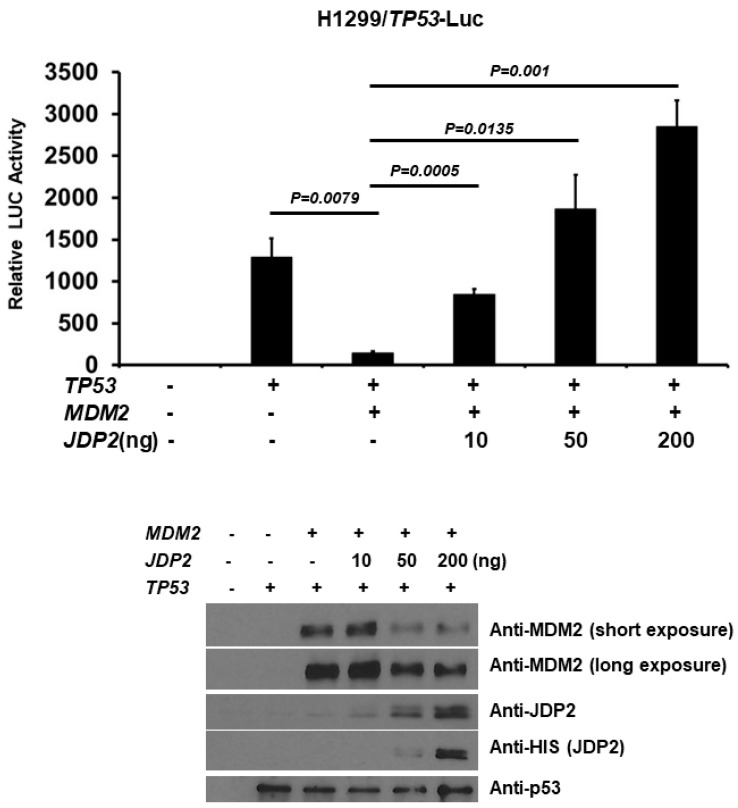
While MDM2 decreases p53 transactivation, JDP2 dose-dependently abolishes MDM2-mediated p53 repression. H1299 cells were co-transfected with the p53 (14X)RE-Luc reporter plasmid and the pRL-TK Renilla control plasmid, along with or without p53, MDM2, and JDP2 (various doses) expression plasmids using Fugene HD transfection reagent. After 48 h post transfection, luciferase activities were measured using the Dual Luciferase Reporter System and normalized to the control Renilla activity. The relative luciferase (LUC) activity was calculated and plotted. A *p* < 0.05 indicates statistical significance between groups. The original western blot figures can be found in Appendix A.

**Figure 7 cancers-16-01000-f007:**
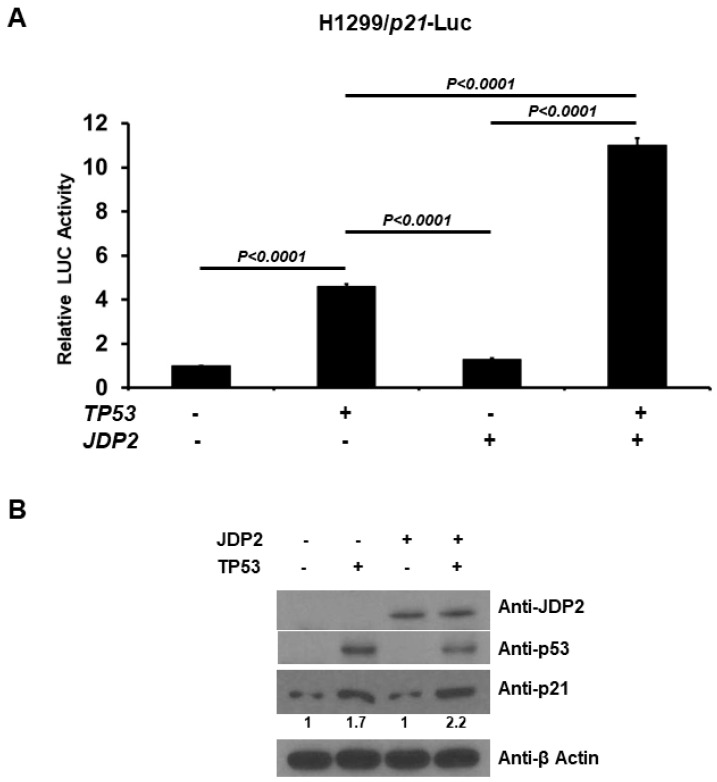
JDP2 increases p21 promoter activity in the presence of p53. (**A**) H1299 cells in a 24-well plate were co-transfected with the p21 promoter(−2347 to +354)-Luc-pGL3 expression plasmid and the pRK-TK Renilla control plasmid, and with or without p53 and JDP2 expression plasmids using the Fugene HD transfection reagent. After 48 h post transfection, luciferase activities were measured using the Dual Luciferase Reporter System and normalized to the control Renilla activity. The relative luciferase (LUC) activity was calculated and plotted. (**B**) H1299 cells were transfected with JDP2 expression plasmid, or p53 expression plasmid, or combination of JDP2 and p53 expression plasmids. After 48 h post transfection, cells were harvested and the whole cell lysates were subjected to anti-JDP2, anti-p53, anti-p21, and anti-β-actin immunoblotting. A *p* < 0.05 indicates statistical significance between groups. The original western blot figures can be found in Appendix A.

## Data Availability

Data is contained within the article or Appendix A.

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
