# Peer review of "Jun Dimerization Protein 2 (JDP2) Increases p53 Transactivation by Decreasing MDM2"

_cancers, 2024, doi:10.3390/cancers16051000_

Round 1

Reviewer 1 Report

Comments and Suggestions for Authors

In their manuscript, Price et al. investigate the impact of the AP-1 protein JDP2 on p53 transcriptional activity. The authors show that overexpression of JDP2 together with p53 in H1299 cells increases the activation of a p53RE-luciferase reporter system, and that JDP2 interacts with p53 through its C-terminus. Finally, the authors suggest that MDM2 downregulation by JDP2 could explain the increased activity of p53.

Overall, the data presented by the authors are robust and the experiments are well controlled. However, the scope of the study is too limited as the authors only use a very artificial model of overexpression to monitor p53 activity without confirming their findings at endogenous protein level. Furthermore, there is insufficient evidence supporting a biological impact.

Major comments:

·         Most of the data rely on overexpression of p53 in a H1299 p53-null cells with an artificial detection system for p53 activity. The authors should demonstrate that JDP2 has a biological impact on endogenous transcriptional activity of p53. For instance, the authors could show how JDP2 overexpression or knock down affects the transcription of p53 target genes in a more physiological context than p53 null cells, such as in MCF7 p53 wt cells.

·         The authors should demonstrate that JDP2 has a biological impact on p53 activity by investigating p53-dependent downstream biological functions.

·         The authors suggest that inhibition of MDM2 is the mechanism for increase p53 activity by JDP2. However, H1299 where most luciferase assay have been performed have a very low endogenous level of MDM2. The authors should demonstrate that JDP2 affects the endogenous protein level of MDM2 in both H1299 and MCF7 cells.

·         The authors suggest that JDP2 decreases the transcription of MDM2. This should be demonstrated (for instance by RT-qPCR).

Minor comments:

·         Some figures are extremely redundant. For instance: Fig. 1B and Fig. 1D could be merged in one figure. Same for Fig.3A and Fig.3B. Fig 3A/3B repeats Fig 1B and 1D. I recommend merging these redundant figures and including additional data in the supplementary section for improved manuscript clarity.

·         The authors mention that “Nur77 has been shown to repress MDM2 transcriptional activity”. this sentence should be rephrased as MDM2 is not a transcription factor and thus does not exhibit any transcriptional activity.

Author Response

Reviewer #1:

In their manuscript, Price et al. investigate the impact of the AP-1 protein JDP2 on p53 transcriptional activity. The authors show that overexpression of JDP2 together with p53 in H1299 cells increases the activation of a p53RE-luciferase reporter system, and that JDP2 interacts with p53 through its C-terminus. Finally, the authors suggest that MDM2 downregulation by JDP2 could explain the increased activity of p53.

Overall, the data presented by the authors are robust and the experiments are well controlled. However, the scope of the study is too limited as the authors only use a very artificial model of overexpression to monitor p53 activity without confirming their findings at endogenous protein level. Furthermore, there is insufficient evidence supporting a biological impact.

Major comments:

  • Most of the data rely on overexpression of p53 in a H1299 p53-null cells with an artificial detection system for p53 activity. The authors should demonstrate that JDP2 has a biological impact on endogenous transcriptional activity of p53. For instance, the authors could show how JDP2 overexpression or knock down affects the transcription of p53 target genes in a more physiological context than p53 null cells, such as in MCF7 p53 wt cells.

Authors’ responses:  Thanks for the reviewer’s suggestion and concerns. Like ATF3, JDP2 is expressed ubiquitously in cells and tissues; however, at normal physiologic conditions, both ATF3 and JDP2 are expressed at low levels. Therefore, we used overexpression system for most of the study. Our long-term goal is to find the physiologic and/or pathophysiologic conditions which are able to increase JDP2 resulting in increasing p53 transactivation and activity in order to combat abnormal cell development. This is our current ongoing project. We added the p21 protein levels in the presence of p53 in Fig. 7B. The results of Fig. 7A and 7B suggest JDP2 increases p53 transactivation and p53-mediated p21 levels.

  • The authors should demonstrate that JDP2 has a biological impact on p53 activity by investigating p53-dependent downstream biological functions.

Authors’ responses:  Thanks for the reviewer’s suggestion and concerns. We added the p21 protein levels in the presence of p53 in Fig. 7B. The results of Fig. 7A and 7B suggest JDP2 increases p53 transactivation and p53-mediated p21 levels.

  • The authors suggest that inhibition of MDM2 is the mechanism for increase p53 activity by JDP2. However, H1299 where most luciferase assay have been performed have a very low endogenous level of MDM2. The authors should demonstrate that JDP2 affects the endogenous protein level of MDM2 in both H1299 and MCF7 cells.

Authors’ responses:  Thanks for the reviewer’s suggestion and concerns. In result 3.5 and Fig. 5, we stated “JDP2 decreases MDM2 level” in both overexpression and endogenous conditions. Both conditions support our finding that JDP2 decreases MDM2 level. While Nur77 has the ability to repress the expression of MDM2 (from the previous published article, reference #48), we do not have data to support whether JDP2 decreases the promoter activity of MDM2. From our new data, it suggests that JDP2 binds MDM2 directly which we have added the data to Fig. 5A. Therefore, we think JDP2 binds MDM2 and then somehow interrupts p53-MDM2 direct interaction, resulting in increased p53 transactivation. Since ATF3 can bind to MDM2 (from another published article, J Biol Chem. 2010 Aug 27; 285(35)), therefore, it is likely that JDP2 can bind to MDM2 which our new data shows. ATF3 and JDP2 share 65% homology in C-terminal domain.

  • The authors suggest that JDP2 decreases the transcription of MDM2. This should be demonstrated (for instance by RT-qPCR).

Authors’ responses:  Thanks for the reviewer’s suggestion and concerns. In result 3.5 and Fig. 5, we stated “JDP2 decreases MDM2 level” in both overexpression and endogenous conditions. Both conditions support our finding that JDP2 decreases MDM2 level. While Nur77 has the ability to repress the expression of MDM2 (from the previous published article, reference #48), we do not have data to support whether JDP2 decreases the promoter activity of MDM2. From our new data, it suggests that JDP2 binds MDM2 directly which we have added the data to Fig. 5A. Therefore, we think JDP2 binds MDM2 and then somehow interrupts p53-MDM2 direct interaction, resulting in increased p53 transactivation. Since ATF3 can bind to MDM2 (from another published article, J Biol Chem. 2010 Aug 27; 285(35)), therefore, it is likely that JDP2 can bind to MDM2 which our new data shows. ATF3 and JDP2 share 65% homology in C-terminal domain.

Minor comments:

  • Some figures are extremely redundant. For instance: Fig. 1B and Fig. 1D could be merged in one figure. Same for Fig.3A and Fig.3B. Fig 3A/3B repeats Fig 1B and 1D. I recommend merging these redundant figures and including additional data in the supplementary section for improved manuscript clarity.

Authors’ responses:  Thanks for the reviewer’s suggestion and concerns. We have decided to use Fig. 1D as new Fig. 1B and Fig. 3B as new Fig. 3A.

  • The authors mention that “Nur77 has been shown to repress MDM2 transcriptional activity”. this sentence should be rephrased as MDM2 is not a transcription factor and thus does not exhibit any transcriptional activity.

Authors’ responses:  Thanks for the reviewer’s suggestion and concerns. Sorry for the confusion. We used the sentence directly from the reference #48. We agree with you that the sentence should be rephrased. We re-write as following: Nur77 represses the expression of MDM2”.

Reviewer 2 Report

Comments and Suggestions for Authors

Thank you for the opportunity to review the manuscript: “Jun dimerization protein 2 (JDP2) increases p53 transactivation by decreasing MDM2". This manuscript presents analyses of the JPD2 protein with the human tumor suppressor protein p53. The topic is generally interesting and the manuscript is well written. However, an improvement would be beneficial before publication and also the mechanism of mdm2 interaction with JDP2 is not clear.

Major points:

1.     The authors show that JPD2 does not increase the transactivation of p53 mutants and claim from this result: 'that JDP2 may not be able to bind to those p53 mutants'. (discussion page 11). However, they did not show immunoprecipitation of these mutants with JDP2 (as is shown in Figure 2 for the wt p53 protein). Therefore, this statement is highly speculative – and most probably wrong. The tested mutants are in the core domain of the p53 protein, which is crucial for its DNA-binding properties –

2.      

3.      so in the case that JDP2 will bind to this part of the p53 protein, it will most likely lead to decreased p53-DNA affinity for p53 DNA and decreased transactivation.

2. The authors showed that the C-terminal part of JDP2 interacts with the p53 wt protein. What part of the p53 protein interacts with JDP2?

3. What are the mechanisms of downregulation of mdm2 downregulation by JDP2? It is direct binding of the JDP2 protein to mdm2 (if yes, which part of the JDP2 protein binds to mdm2?) or is it competition for the N-terminal part of the p53 protein?

4. Are the JDP2 and p53 protein both expressed in some human tissues?

Minor points:

Please correct typos: e.g.  “enhance” (title 3.3), “Hosuton” (chapter 2.2) etc.

More recent citations would be welcome.

Comments on the Quality of English Language

Please correct typos: e.g.  “enhance” (title 3.3), “Hosuton” (chapter 2.2) etc.

Round 2

Reviewer 1 Report

Comments and Suggestions for Authors

No additional comments

Reviewer 2 Report

Comments and Suggestions for Authors

The authors have improved the manuscript.